# How different autonomous vehicle presentation influences its acceptance: Is a communal car better than agentic one?

**Konrad Hryniewicz**[1]*, **Tomasz Grzegorczyk**[2]

**1** SWPS University of Social Sciences and Humanities, Warsaw, Poland, **2** Poznan University of Business and Economics, Poznan, Poland

* khryniewicz2@st.swps.edu.pl

## Abstract

Public acceptance of autonomous vehicles (AVs) is still questionable. Nevertheless, it can be influenced by proper communication strategy. Therefore, our research focuses on (1) the type of information concerning AVs that consumers seek and (2) how to communicate this technology in order to increase its acceptance. In the first study (N = 711) topic modeling showed that the most sought for information concern the communion and the agency of AVs. In the second, experimental study (N = 303) we measured the participants' fear and goal-orientation in relation to AVs. Then, after the manipulation of the AV advertisement (imbued with communal vs agentic content), technology acceptance components (perceived ease of use, perceived usefulness and behavioral intention) were verified. The comparative analysis of the structural model estimates showed that the both participants' fear and goal-orientation in relation to AVs were associated much more with the acceptance components of the communal AV rather than the agentic one. Therefore, people want to know both whether AVs are communal and agentic, but they are more prone to accept a communal AV than agentic one.

## Introduction

The development of artificial intelligence and real-time data processing technologies have enabled the introduction of autonomous vehicles. Autonomous vehicles (AVs), also known as self-driving cars or driverless cars are vehicles that can drive without the aid of a human operator [1]. The Society of Automotive Engineers (SAE) classifies vehicle automation into six levels [2]. Vehicles with conditional automation (Level 3), high automation (Level 4), and full automation (Level 5) are regarded as "AVs", while the term "self-driving vehicles" (SDVs) is reserved to level 5 AVs [3]. Currently, vehicles are only partly autonomous (level 2), capable of, for instance, autonomous parking. However, vehicle companies, including Ford, Honda, Toyota, Nissan, Volvo, Hyundai, Daimler, Fiat-Chrysler, and BMW increase investments in the development of autonomous vehicles [4]. Naturally, AVs are regarded as one of the fundamental disruptors in the upcoming technological revolution [1, 5].

**Data Availability Statement:** All files and data are available on Open Science Framework (DOI: 10. 17605/OSF.IO/3URZG).

**Funding:** The authors received no specific funding for this work.

**Competing interests:** The authors have declared that no competing interests exist.

AVs have many advantages over traditional cars. Firstly, AVs may improve roadway safety due to the elimination of human error which is the cause of 93% crashes in the U.S. [6, 7]. Secondly, due to better route planning and more efficient operation, AVs are predicted to reduce road congestion, fuel emission and fuel economy [7–9], as well as to improve pavement stability [10, 11]. Thirdly, what may be important for consumers is the ability of AVs to save time and provide comfort and leisure, as they allow the passengers to engage in non-driving activities [12–14]. Finally, AVs enable comfortable traveling for people unfit for independent driving, e.g., the elderly and people with disabilities [15]. However, introduction of AVs is connected with multiple challenges such as safety, legal liability, ethical questions and regulatory issues, which result in consumers' fear towards this tech [16–18]. AVs also are a challenge to the traditional role of drivers and driving pleasure [19]. Even though the benefits of AVs seem to significantly outweigh the risks associated with them, consumer acceptance of this technology is still uncertain. It is, however, crucial for its diffusion and commercial success [20–22]. While many studies concerning attitudes towards AVs were conducted, the results are mixed. Some found that the positive attitude prevails (e.g. [18, 23]), while other show the contrary (e.g. [17, 24]).

Researchers have also been studying the factors which influence consumer acceptance of AVs. By adapting and creating models it is possible to understand and predict consumer behavior in the AV context. Multiple theories have been used for this purpose. However, none of the previous studies draw upon the dual perspective model of agency and communion (DPM-AC) [25] which shows that the assessment of social objects depends mainly on safety concerns and personal goals. Moreover, Liu et al. [26] underline that especially psychological determinants of AV acceptance remain insufficiently investigated and largely unknown [27]. Also, Zhang et al. [13] noticed that on average only about 50% of variance in acceptance was explained by existing models (e.g. [28–30]). We aim to fill this gap by not only providing new external variables which predict AVs acceptance, but also by testing novel conditions influencing acceptance (AV advertisement imbued with specific content). Moreover, as the large survey on 27 thousand EU citizens showed, cultural differences play a significant role in the acceptance of autonomous agents [31]. Meanwhile, no such study was conducted in Poland.

By giving an AV either agentic or communal traits in accordance with DPM-AC and testing the differences in consumer acceptance it is possible to identify the appropriate way to design AVs and formulate related communication strategies (both for governing authorities and private companies). This should allow to positively influence attitudes towards AVs which is crucial for their long-term diffusion and success. The research dedicated to this issue is still scarce [3, 32]. Therefore, the aim of our study is to determine the type of information concerning AVs that people seek and identify how different AV technology presentation modes (advertisements imbued with agentic or communal content) influence AV acceptance among people who are either afraid of AVs or aim to achieve their goals thanks to them.

## Acceptance of autonomous vehicles

Technology acceptance is understood as the degree to which an individual is likely to accept and intends to use a particular technology. Citizen acceptance (e.g. general public response to a particular technology) should be distinguished from consumer or user acceptance (e.g. the intention to purchase and use a particular technology) [30]. Previous research on the acceptance of AVs can also be divided into the research based on the behavior theories and without them. The latter are mostly descriptive studies (e.g. [23, 33, 34]) which are mainly focused on citizen acceptance.

Many of them concentrate on demographic factors. It has been confirmed that younger male drivers had a more positive attitude towards AVs and were more willing to buy one [23, 34–37]. Moreover, such factors like higher income [35], living in an urban area [38], tech-savvy and also involvement in more crashes [39] positively affect the attitudes towards AVs [13]. A meta-analysis of studies not based on behavior theories showed that the acceptance of AVs was explained by safety, performance-to-price value, mobility, travel time, symbolic value, and environmental friendliness [40]. The role of legal liability and regulation issues of AVs was also confirmed [26].

According to a meta-analysis of AV acceptance studies [40] the most cited behavior theories were respectively: Technology Acceptance Model (TAM, [41]), Theory of Planned Behavior (TPB, [42]), Unified Theory of Acceptance and Use of Technology (UTAUT, [43]), Diffusion of Innovation Model (DIM), [20, 44]), Theory of Reasoned Action (TRA), [45]). Researchers use these models in order to determine which external factors explain technology acceptance and resistance [46, 47]. Authors of AV's acceptance models often adapt the original models and incorporate novel factors. Among the most cited factors were respectively: perceived ease of use, perceived usefulness, trust, attitude, social norm, perceived risk and compatibility [40]. Such research is recently rapidly gaining in numbers (e.g. [1, 26, 29, 30, 34, 48, 49]).

As the number of studies rises, researchers begin to focus on more detailed situations, technologies and respondents, e.g. ride-sharing [50–53], long-distance mode choices [50], partial automation after brief exposure [54], autonomous vehicles in connection with alternative fuel technology [55] or connected driving [56, 57], the effect of direct experience of autonomous drive [3], blind respondents [58], influence of children mobility [59], autonomous delivery vehicles [60] and AV communication strategy [32].

The research on the AV communication strategy is very limited and the research on acceptance of AV would be best suited to draw from when dwelling into this matter. However, one such exception is Myrick et al.'s [32] study on the effect of the name frames for AVs (e.g. a driverless car) and celebrity endorsement on the public perception of risk, benefit and intentions.

## Technology acceptance model

As mentioned, TAM is the most cited behavior theory in AV acceptance research [40]. It was developed by Davis [41] to predict information technology acceptance. Since that time many technology acceptance models based on this theory have been developed and validated [61] in various technology contexts [62]. Originally TAM was embedded in TRA [63] which developed into the Theory of Planned Behavior [42].

In this article we use perceived ease of use, and perceived usefulness to predict AV's consumer acceptance understood as intention to use in accordance with Davis's [41] seminal paper about TAM. As was the case in the past, intention construct is considered a proximal factor of actual behavior [64, 65] and is especially useful in terms of measuring the acceptance of technologies at an early stage of development [48]. Davis defined perceived ease of use as a personal belief that using a technology is free of effort, while perceived usefulness is understood as a belief that using a technology will enhance personal performance [41]. Davis provided explanation and confirmation that the 1) perceived ease of use and 2) perceived usefulness have strong impact on personal 3) intention to use technology, while perceived ease of use also positively influences perceived usefulness [41]. In this article we assumed that these three variables are the main components of technology acceptance.

However, previous research based on TAM that delved into AVs showed mixed results. Only a few studies (e.g. [66, 67]) confirmed that all those relationships are significant, while

most of the studies provided confirmation for only some of them. Xu et al. [29] showed that perceived ease of use and perceived usefulness positively influence the intention to use AVs, but they did not test the relationship between those two variables. Moreover, Choi and Ji [48] reported that perceived ease of use did not influence the perceived usefulness, while both those factors positively impacted intention to use AVs. Moreover, Hein et al. [68] as well as Lee et al. [4] and Hegner et al. [69] showed that the perceived ease of use did not influence the intention to use (contrary to the effect of perceived usefulness), while it influenced positively perceived usefulness and intention to use. Interestingly, in the case of level 3 AVs, it was reported that it was the ease of use which predicted the intention to use and not perceived usefulness [30]. All in all, it was the perceived usefulness which had most often the strongest influence on intention to use AVs [40].

We agree with Lee et al.'s [4] conclusion that studies of AVs based on technology acceptance models should start by examining TAM's basic assumptions. We are convinced that the TAM model in this basic form is simple, well suited to both study the AV technology, and to test acceptance in different experimental conditions. In this article, we want to introduce new factors (observer's and agent's perspective) which are fundamental perspectives in social cognition [25, 70]. We predict that these personal dispositions are not only highly related with TAM components but these relations are also heavily influenced by how AV technology is presented.

## Agency and communion

Bakan [71] was among the first researchers who wrote about a human being in the context of agency and communion. According to his dual vision of human existence, every person is a pursuer of his or her goals (he or she is agentic), but also everyone is a member of a community and participates in various social relations (he or she is communal). Currently, agency and communion are the fundamental dimensions known as the Big Two [25], which are theoretical concepts used to explain human perception of self and others in terms of valuation of behavior and traits [72–74], liking, respect [75], and also attitudes toward products [76, 77].

Researchers agree that the dimension of communion refers to the initiating and maintaining safe and satisfying social relations, and the dimension of agency refers to the implementation of tasks and achieving personal goals [25, 78]. The former contains such characteristics as: warm, nice, helpful, cooperative, and trustworthy (along with their opposites), while the latter embodies such characteristics as: efficient, competent, active, persevering, and energetic (with their opposites). These content dimensions are manifested in natural language [79], spontaneous self-representation [80], and information processing [81].

Nevertheless, these fundamental dimensions do not have equal importance in social cognition. Researchers show that communal and agentic content plays different roles in managing behavior [25, 72, 82, 83]. It turns out that communal content has sometimes greater importance for people's behavior than agentic content, but sometimes this pattern is inverted.

Communal content plays a more important role than agency in forming impressions of others [84]. Communal content in a person is a better predictor of his or her likeability than agentic content [75]. People also recognize and categorize communal words faster than agentic ones [85]. They are more interested in communal content as it serves as it allows them to more accurately predict the behaviours of both real people and fictional characters [86]. Studies in the area of advertising effectiveness have shown that a communal endorser causes greater liking and more positive attitudes towards the advertisement and the brand [76]. Such advertisements also form positive attitudes towards offered sports services and a sense of greater self-

efficacy while using them, which in turn leads to the intention to exercise [77]. Recent study also showed that recipient's communal orientation is correlated with attitudes towards advertisements that express communal traits [87]. All aforementioned phenomena are called the communion over agency effect [84–86].

But sometimes in social cognition, an opposite effect occurs when agentic content achieves greater importance than communal content. That pattern is called the agency over communion effect. It appears that when a person is goal-oriented, then he/she assigns greater importance to agency, and as well as when personal outcomes depend on someone else's behavior, then his/her agency is more important than communion [72, 81, 88]. For example, in business-oriented organizations, the employee's bonuses, his or her goals and achievements depend on the executive qualities of their superior. Therefore, to the employees of such companies, the agency of the leader is more important than his communion (relation of interdependence). Similar interdependence effect appears in the advertising context when the product promises to help the consumer achieve personal goals [87, 89]. Even though these results do not pertain to technology acceptance, they reveal something about the specific nature of the agentic and communal content for social cognition.

## Dual perspective model of agency and communion

We introduce the dual perspective model of agency and communion (DPM-AC) which integrates and explains a wide spectrum of aforementioned phenomena [25]. We tie this theory with self-congruity effect [90] for a better understanding of evaluation processes in the technology acceptance context. Using these tools, we may create a framework for making predictions regarding the acceptance of AVs that are characterized by agentic or communal traits.

DPM-AC refers to the claim that agency and communion have a slightly different meaning for a person who rates someone else's behavior (observer's perspective) compared to the person oriented towards an intended action (agent's perspective) [25, 91]. The model provides an explanation for the differences in terms of the behavior of the observer interpreting the behavior of other people and agent interpreting his own actions. These mindset differences have crucial impact on human perception and behavior. This model indicates that agency and communion are the contents of social reality which have a certain adaptive value for the agents and observers alike [92, 93].

Abele and Wojciszke claim [25] that communal traits in other people are beneficial for the observer because they inform him or her about the intentions (good or bad) of the observed person [70, 74]. The knowledge that another person (a psychologist) is communal, for example, that he or she is warm and moral, allows the observer to make a positive inference and assume good intentions. On the other hand, agentic traits (e.g. a professor's wisdom) are beneficial for the person pursuing his or her goal (e.g. college graduation) [25, 94].

The first prediction of our research model concerns the perspective of the observer who finds communal traits in others to be more significant than agentic ones. In this perspective, a person prioritizes safety and simultaneously expects benefits which result from the behavior of another person. The second prediction concerns the perspective of the agent to whom the agentic traits are more significant than the communal ones. The agent wants to efficiently pursue his goals, which results in the attachment of greater importance to agentic traits than to the communal ones [25]. In our research, we treat these perspectives as quantitative dimensions, as a recent study showed that people differ in propensities to take agent and recipient perspectives [91].

## A congruence between technology presentation and fundamental perspectives

Perception patterns of other people and perception patterns of technology may be quite similar because people are able to attribute mental states and traits to non-social objects and other non-human beings [89, 95–98]. Thus, we introduce a self-congruity effect to better understand valuation process which results in positive attitude towards product [90, 99], brand trust and loyalty [100, 101], and purchase intention [102–104]. Technology acceptance of the product may result from the congruity between main perspectives and a technology presentation that contains content which is important for its users. A particular form of technology presentation, for example an advertisement, catalogue description or article, saturated with agentic and/or a communal content can be in tune with the basic needs important for a given observer and/or the agent perspective. Depending on the type of content in the AV technology presentation, the presented technology will be valuated, and accepted [90, 102] through particular perspectives [25, 72]. A person who is more concern-oriented (observer's perspective) will accept a car presented in a communal manner (in a safe, family context), because in this perspective the communal content is evaluated more positively than the agentic one. On the other hand, a goal-oriented person (agent's perspective) who wants to move quickly and efficiently around the city will accept cars presented in an agentic manner (a combination of perfect technology and design), because in this perspective the agentic content is valuated more positively than the communal one. The resulting attitudes towards a given technology, as well as intention to use it, are the result of the motivational state stemming from the need to strengthen self-esteem and maintain behavioral consistency [90, 105].

Metaphorically speaking, aforementioned perspectives are a lens through which people evaluate different objects in terms of meeting their personal concerns and needs [106, 107]. Therefore, we predict that the AV technology will be accepted if its presentation is saturated with a type of content that is congruent to them. To the best of our knowledge, there is no study that used these theories to predict technology acceptance.

## Hypothesis

The general objective of our investigation is to (1) determine the type of content concerning AVs that people seek and (2) identify how different AV technology presentation influence AV acceptance among people who are either afraid of AVs (observer's perspective) or aim to achieve their goals thanks to them (agent's perspective). By equivalently transposing the predictions of the DPM-AC and self-congruity effect into the context of technology perception and technology acceptance models, we formulate testable predictions about the emergence of technology acceptance in given circumstances.

The first prediction is related to the basic needs predicted by DPM-AC [25]. According to the agents's perspective, people want to know how well an AV technology works. The attachment of greater importance to agentic traits of the car originates from the need to pursue goals efficiently. Nevertheless, according to the observer's perspective, people want to know how communal an AV is, as it allows them to avoid harm and gain benefits. Therefore, we hypothesize that:

H1: People want to know how agentic an AV technology is and how communal it is.

AV technology advertisement imbued with more communal content is valued from the standpoint of the observer. In this perspective, the technology's recipient wants to avoid harm and gain benefits. Moreover, the information about AV's communion (technology's safety) is more important than the agentic one (how well it works). The more the recipient identifies

with the observer's perspective, the higher the acceptance of an AV technology which is saturated with communal content rather than agentic one. We hypothesize that:

H2: The observer's perspective more positively influences the TAM's components of a communal AV than the TAM's components of an agentic AV.

By contrast, an AV technology advertisement imbued with more agentic characteristics is positively related to the recipient's actions towards his/her goals. In this perspective, the agentic traits of a technology (e.g. a well-functioning technology) are more important to the individual than the communal ones (safety and secure). The more the recipient identifies with pursuing different goals [42, 63, 78] by AV, the higher acceptance he or she assigns to an agentic AV technology. Therefore, the following hypothesis will be tested:

H3: The agent's perspective more positively influences the TAM's components of a agentic AV than the TAM's components of a communal AV.

In the course of two studies, we verified the hypotheses concerning the information needs and the acceptance of the AV technology. The research model is presented in Fig 1.

## Study 1

At this stage, we aimed to confirm the consumers' expectations regarding agentic (efficient technology) and communal (safe technology) AV technologies according to the agent's and observer's perspectives. Thus, we gather and investigate data to verify H1.

### Method

At the beginning of the survey, the participants gave their informed consent to take part in the study (in accord with the APA 3.10 standard [108]). The research was carried out using the Lime Survey program [109] on Facebook groups which united students. In the first step of the survey, participants put their demographic information, and then in the second step, we asked them 5 open-ended questions:

1. What would you like to see in advertisements about transport services in which autonomous cars run?

2. What kind of message that advertisement would convey?

3. What would you like to happen in an advertisement or presentation of such a car?

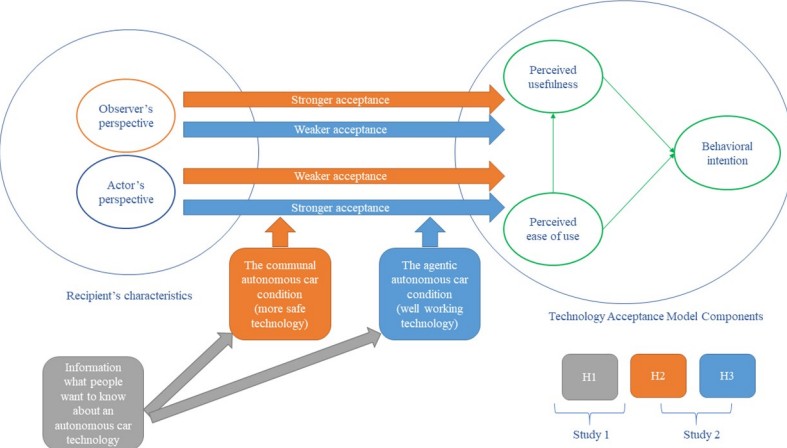

**Fig 1. Research model.**

4. What would you like to see or find out about the transport based on self-driving cars?

5. What values or ideas would be conveyed in a self-driving car advertisement?

Next to these questions, participants entered their answers. All answers were translated from Polish to English.

710 participants (179 male) took part in the study. Age M = 24.16; SD = 7.29. Basic education had N = 44 participants, secondary education N = 416, and N = higher education 250. To analyze gathered information in the open-ended questions, Natural Language Processing was performed in Orange 3 Biolab Studio [110, 111]. A tokenization procedure regexp /w+, which matches only words, was used as well as a normalization words procedure by Word-Net Lemmantizer. WordNet Lemmantizer is a huge word database for the English language aiming to establish structure and relationships between words. It offers lemmatization capabilities as well and is one of the most commonly used lemmatizers in text mining calculations [112, 113]. Preprocessing text procedure resulted in 711 documents, 7797 total word tokens, and 111 token types. To establish the AV-related topics raised by participants, the topic modeling procedure was performed by Latent Dirichlet Allocation algorithm [114, 115]. The topic modeling method is an unsupervised analysis which uncovers topics from a set of documents. A topic is determined as a distribution over a fixed vocabulary. Latent Dirichlet Allocation analyzes the words in each document and calculates the probability distribution between the words in the document, and the latent structure of topics. Based on the theory DPM–AC [25], two latent topics were chosen (in accordance with observer's vs agent's perspective).

## Results

Topic modeling results showed that topic 1 was dominated by the following words: safe, driving, safety, road, work, family, security, comfort, environment, beautiful, situation, child, appearance, look, passenger, happy, etc. Topic 2 was dominated by the following words: advantage, people, information, use, good, way, know, transport, important, driving, operation, system, nice, service, product, solution, etc. The results are shown in Figs 2 and 3. All words and their topic weights are given in S1 Table.

Conducted analysis showed that the participants raised two different topics about the AV. Topic 1 was more related to the communal content (in congruence with observer's perspective), and topic 2 was more related to the agentic content (in line with agent's perspective). Due to the research and statistical method, it is difficult to draw unambiguous conclusions concerning the specific nature of AVs expectations in relation with agent's and observer's perspectives. However, it is quite clear that people want information both about safety and how AV technology works.

## Study 2

In this stage, we created and tested two different technology advertisements (agentic vs communal AV) in a pilot study. Next, we manipulated them in an experiment. The experiment was designed to verify relations between two perspectives and TAM components in the two different technology advertisement conditions. At this stage of our investigation we tested H2 and H3.

## Pilot study

This pilot study was aimed at creating two different advertisements for AV technology. Two separate personalities, one agentic and the other communal were created for the AVs [89]. In

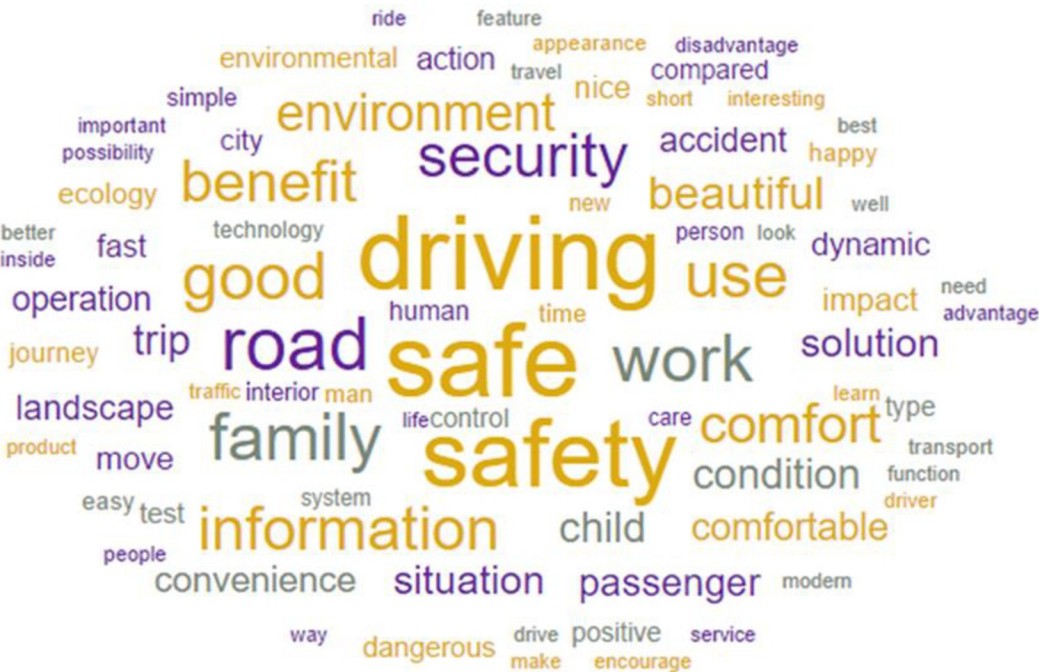

**Fig 2. Words appeared in the topic 1.** The bigger word the more frequently it occurs.

these descriptions, the created CRUISE AV talked about himself. Such an anthropomorphiza-tion allowed to create two versions of the advertisement in which the AV CRUISE had either agentic or communal characteristics. These research materials will be used in further experi-ments with the aim to verify H2 and H3.

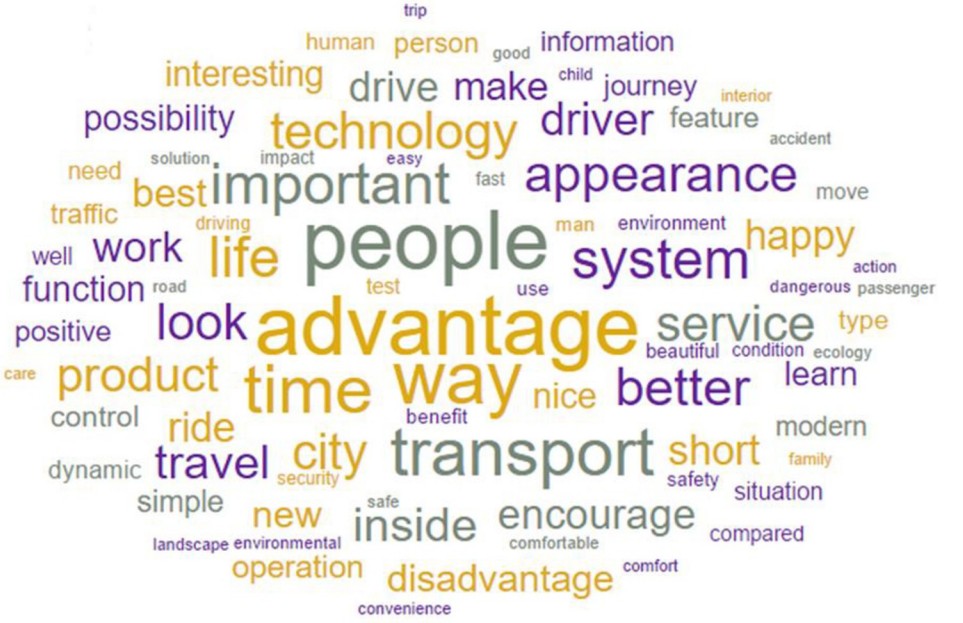

**Fig 3. Words appeared in the topic 2.** The bigger word the more frequently it occurs.

**Method.**   At the beginning of the survey, the participants gave their informed consent to take part in the study (in accord with the APA 3.10 standard [108]). In the on-line survey conducted on adults interested in the car market (traders and customers), a poster displaying the AV (named CRUISE) and its catalog description were presented. The respondents completed an on-line survey and were randomly assigned to the condition of either agentic or communal car advertisement. At the beginning of the study, the participants had to familiarize themselves with the image of the product and its catalog description. The presented car is depicted in the Fig 4.

Two examples of information presented to the respondents on a poster along with the product are shown below.

Agentic car condition: "I am the autonomous car CRUISE, an intelligent invention designed for efficient and fast driving in the city and on the highway. I work very surely and meticulously, which guarantees fast and accurate commute between point A and point B. My precise sensors, clever and reliable measuring electronics constantly monitor all road events. Thanks to this, I react extremely effectively to even the smallest changes in traffic conditions. I do not quit on steering until I make an uncompromising and complete clarity of all situations on my way. I consistently strive to provide excellent driving performance through rapid analysis of information via artificial intelligence, processors, and sonars. I analyze specifically all the details pertaining to the commute. When I arrive at the destination, I turn into the mode of constant vigilance and traffic monitoring".

Communal car condition: "I am an autonomous car and my name is CRUISE. Together with the family and the dog, we surround ourselves with care and help. We get up to work and school together, and then we go back home and rest. We live and support each other. We accompany each other in these less and more important moments of our everyday lives. Me, mom, dad and grandma sometimes go and pick up canvasses for grandpa's paintings. Together, we read fairy tales to the daughter and often sing without hesitation when we go on trips outside the city. We cried when eight puppies were born while we were on our way to the

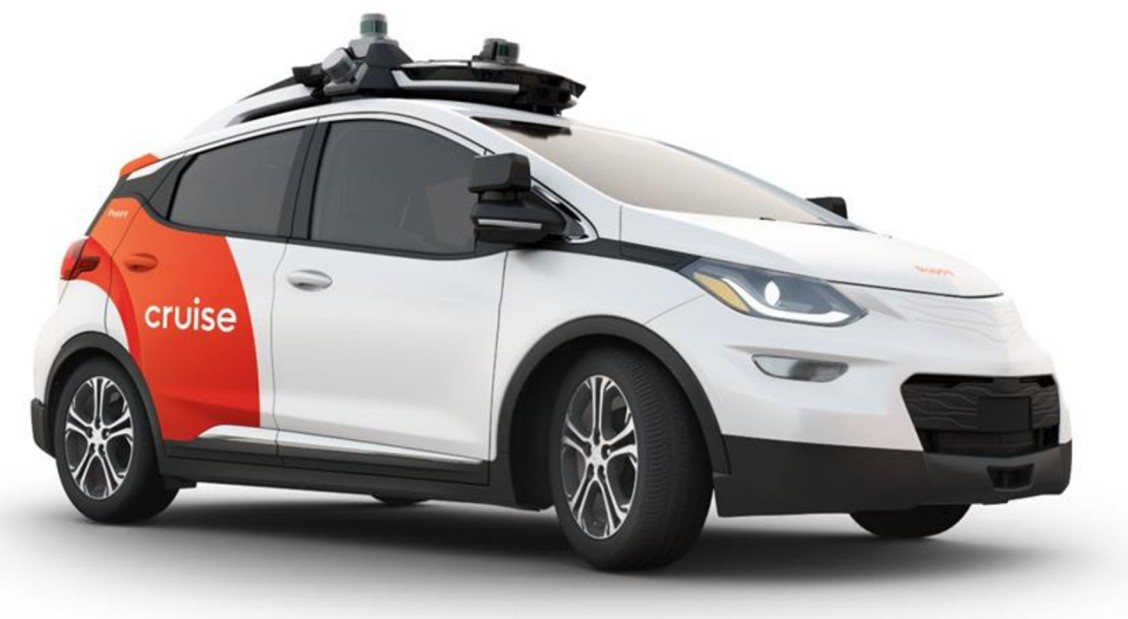

**Fig 4. CRUISE car which was presented along with communal or agentic description (photo can be retrieved form [116]).**

vet, and we also celebrated when dad won the children's book competition and mom finished a culinary course. Well-being and having a good time do not end when we are on the move. We relax whenever we are close to each other. Being in constant motion, we all care about each other's safety. Safety is very important for our family".

In the next step of the survey, the participants were asked to assess the previously presented product on the widely used adjective scale of agency (10 items: efficient, confident, competent, never gives up, smart, leader type, effective, dominant, intelligent, copes well under pressure), and communion (10 items: trustworthy, caring, acts fairly, kind, just, cordial, considerate, empathic, reliable, warm) [117, 118].

The recipients responded on a seven-point Likert scale (1 = Definitely doesn't suit and 5 = Definitely suits). They also rated the level of car CRUISE anthropomorphization on the Likert scale (1 = Definitely not characterized/ 5 = Definitely characterized). This measure was based on 4 items: The CRUISE car behaved like a human; The CRUISE car resembled a human; The CRUISE car was very human; The car had human characteristics. Reliability of the measure for agency, communion, and anthropomorphization level was $\alpha = 0.81$, $\alpha = 0.84$, $\alpha = 0.87$, respectively.

**Results.** $N = 117$ (65 male) participants took part in the study. Their age was M = 29.75; SD = 4.51. To analyze results, ANOVA was conducted. It showed a significant interaction effect between groups (agentic car N = 58 vs communal car N = 59) and ratings of the car in term of agency and communion $F(1.115) = 132.62$; $p < 0.001$, $\eta^2 = 0.54$. Within-group analysis showed that the rating of the agency was higher in the agentic car group than the communal one $F(1.115) = 50.31$; $p < 0.001$, $\eta^2 = 0.30$. Analysis showed also that the rating of the communion was higher in the communal car group than agentic one $F(1.115) = 53.18$; $p < 0.001$, $\eta^2 = 0.32$. Between-group analysis showed that in the agentic car group, the rating of the agency was higher than communion $F(1.115) = 36.15$; $p < 0.001$, $\eta^2 = 0.24$, and in the communal car group, this pattern was inverted $F(1.115) = 105.92$; $p < 0.001$, $\eta^2 = 0.48$. Results are shown in the Fig 5. Another ANOVA showed that the car anthropomorphization ratings were higher in the communal car group $M = 4.00$; $SD = 0.92$ than agentic one $M = 3.19$; $SD = 0.95$, $F(1.115) = 21.88$; $p < 0.001$, $\eta^2 = 0.16$.

Analysis showed that prepared research materials were characterized by desired features. The agentic AV was perceived much more agentic, and the communal AV was perceived much more communal. Analysis showed also that the communal car (vs agentic car) was also perceived much more like a human being (anthropomorphization measure). Despite significant differences between groups in terms of anthropomorphization, this difference was smaller than difference between groups in term of agency and communion, respectively effect sizes were $\eta^2 = 0.16$ vs $\eta^2 = 0.30$ vs $\eta^2 = 0.32$. To summarize, the first car was perceived as functional and efficient (assigned high agency) and the second one was perceived as much safer (assigned high communion).

## Main experiment

We designed an experiment to verify H2 and H3. Through the course of the survey, participants were first measured in terms of the agent's and observer's perspectives. After that, they were randomly assigned to agentic or communal autonomous CRUISE car advertisements. In the next step, they rated presented AV technology in terms of the TAM components. The analysis of results is divided into two sections. In the first one, we presented differences between groups (agentic vs communal car) in terms of the TAM's components and dual perspectives measurements, constructs validity statistics, and measurement invariance analysis. In the

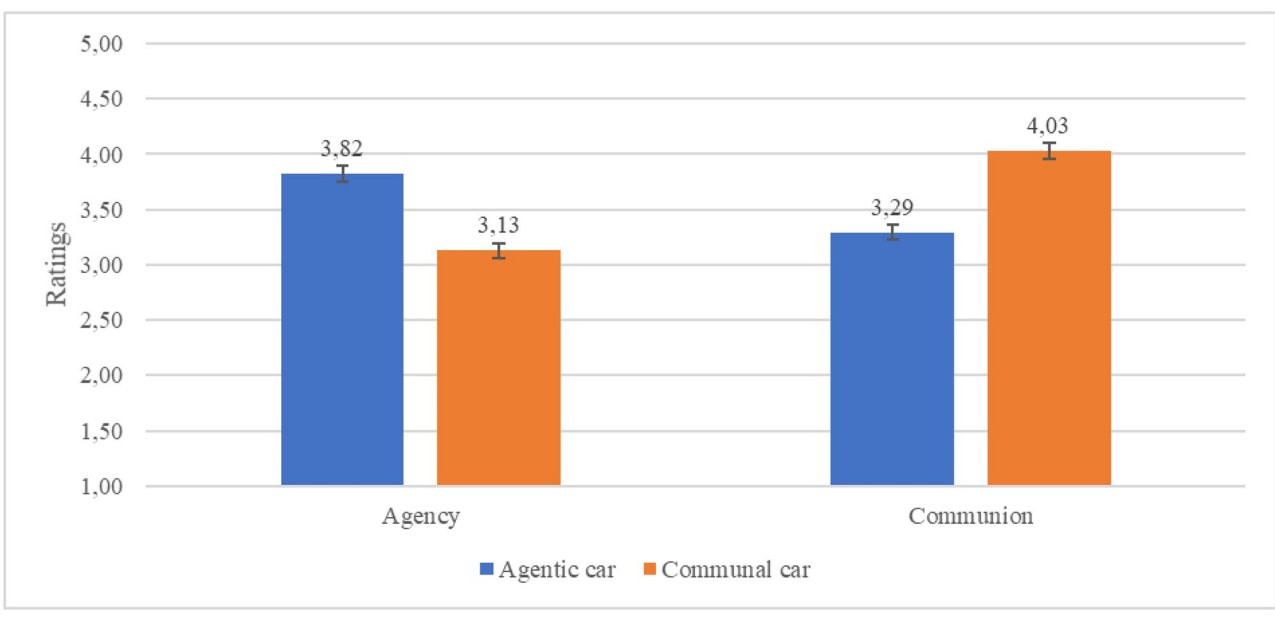

**Fig 5. Agency and communion ratings of the agentic and communal CRUISE car.**

second section, we presented how measured agent's and observer's perspectives predict TAM's components in two experimental groups (agentic vs communal AV).

**Method.** At the beginning of the survey, the participants gave their informed consent to take part in the study (in accord with the APA 3.10 standard [108]). The research was carried out using the Lime Survey program [109] on Facebook groups which unite automobile traders and workers. N = 303 participants (173 female) took part in the study, in age M = 27.52; SD = 5.87. Basic education had N = 2 participants, secondary education N = 130, and higher education N = 171. 277 participants had a driving license (N = 22 with no license). All participants had driven a car. All the items in each survey step were randomized. In the first step, participants filled out the demographic information. In the second one, they were informed about the AV concept via a short description: „The following survey concerns the perception of autonomous cars which are driven thanks to artificial intelligence systems. You don't have to do anything because the car is controlled by a computer. Please refer to the following statements. There are no right or wrong answers. Only personal opinion about how you feel, perceive and evaluate the world and yourself matters". In this step, participants had to assess themselves on the observer and agent perspective scale. All these items are presented in Table 1. Answers were recorded on a five-degree Likert scale (1 = Definitely doesn't suit me, 5 = Definitely suits me). In the third step, the participants were randomly assigned to one of the two technology advertisement conditions (agentic vs communal). In the fourth step of the study, the participants assessed the autonomous CRUISE car technology on 3 dimensions of technology acceptance model combined with a five-degree Likert scale (from 1 = I definitely do not agree/ to 5 = I Definitely agree). TAM measures are presented also in Table 1.

## Analysis 1. Differences between groups (agentic vs communal AV) in terms of the technology acceptance and dual perspectives measurements

**Model fit.** To assess differences between groups (agentic vs communal car) in terms of the technology acceptance and dual perspectives measurements the PLS–SEM was conducted in

**Table 1. Measurements used in the study.**

| Scale and item id | Item |
|---|---|
| PEU 1 | Using the CRUISE autonomous car does not require much effort. |
| PEU 2 | Driving and moving by autonomous CRUISE cars is easy. |
| PEU 3 | The CRUISE autonomous vehicles are easy to use and everyone will understand them. |
| PEU 4 | There is nothing difficult in travelling with a CRUISE car. |
| PU 1 | The CRUISE autonomous car is a very useful thing. |
| PU 2 | The CRUISE self-driving cars can do a lot of good. |
| PU 3 | The autonomous CRUISE car will become a symbol of utility. |
| PU 4 | The usefulness of the autonomous CRUISE car will be widespread. |
| BI 1 | I would love to drive a CRUISE autonomous car. |
| BI 2 | If self-driving CRUISE cars appear, I will try them immediately. |
| BI 3 | I think I could spend my money on the CRUISE autonomous car. |
| BI 4 | I intend to use the CRUISE autonomous car. |
| O 1 | I am very afraid that an autonomous car may cause an accident when I will be driving it. |
| O 2 | I feel anxiety that the car can be driven by a computer and not by a human. |
| O 3 | I do not like situation that I may not have control over how and where the autonomous car is going. |
| O 4 | Somehow, I'm scared of traveling by a self-driving autonomous car. |
| O 5 | I would feel uncertainty when driving a car driven by a computer and its artificial intelligence. |
| A 1 | In my everyday life, a car controlled by a computer would bring me many benefits. |
| A 2 | A car controlled by artificial intelligence could help run a lot of errands. |
| A 3 | If I had an autonomous car, I could do many new things. |
| A 4 | An autonomously computer-driven car would save my time. |
| A 5 | Thanks to traveling by autonomous car, I could work more efficiently. |
| A 6 | A self-driving car would help me out with many of my problems. |

PEU = Perceived ease of use, PU = Perceived usefulness, BI = Behavioral intention, O = Observer's Perspective, A = Agent's perspective.

WarpPLS 7.0 software [119, 120]. In congruence with Dijkstra and Hensler [121] the Consistent PLS algorithm was selected to establish path estimates in the proposed model. The fit statistics showed in Table 2 inform that the tested model had small collinearity within path model (AVIF) and measurement model (AFVIF) [122, 123], lack of predictive power (GoF) [124]. The path signs (SPR) and path values (SSR) in the model were similar to the signs and values of the independent zero-order correlations between the tested variables [124, 125]. The analysis also displayed a very good data fit to the measurement model of latent variables (SRMR, SMAR, $\chi^2$). All consistent reliabilities were acceptable $\rho A > 0.88$ [121, 126]. Detailed information about consistent reliabilities are showed in Fig 7.

**Path analysis results.** Analysis of the path model showed that the group did not influence technology acceptance and dual perspective measurements. The results are shown in Fig 6.

**Construct validity and measurement invariance.** To check construct validity a heterotrait-monotrait (HTMT) ratios [127], an Average Variance Extracted coefficients [128], and zero-order correlations were calculated. Obtained results presented in Table 3 were found to be far better than acceptable threshold levels [119]. Invariance analysis showed that the factor loadings were similar in both conditions t < 1.36. These results are shown in Table 4.

**Table 2. Model fit and quality indices.**

| Statistics | Coefficient |
|---|---|
| AVIF | 0.00 |
| AFVIF | 1.58 |
| GoF | 0.04 |
| SPR | 1.00 |
| SSR | 1.00 |
| SRMR | 0.08 |
| SMAR | 0.06 |
| $\chi^2$ | 0.37*** |

AVIF = Average Variance Inflation Factor (accepted if AVIF < = 5.00, ideally AVIF < = 3.30); AFVIF = Average Full Variance Inflation Factor (accepted if AVIF < = 5.00, ideally AVIF < = 3.30); GoF = Goodness of Fit (low if GoF > = 0.10, moderate if GoF > = 0.25, high if GoF > = 0.36); SPR = Simpson's Paradox Ratio (accepted if SPR > = 0.70, ideally SPR = 1.00); SSR = Statistical Suppression Ratio (accepted if SSR > = 0.70, ideally SSR = 1.00); SRMR = Standardized Root Mean Squared Residual (accepted if SRMR < 0.10); SMAR = Standardized Mean Absolute Residual (accepted if SMAR < = 0.10); $\chi^2$ = Chi Square.

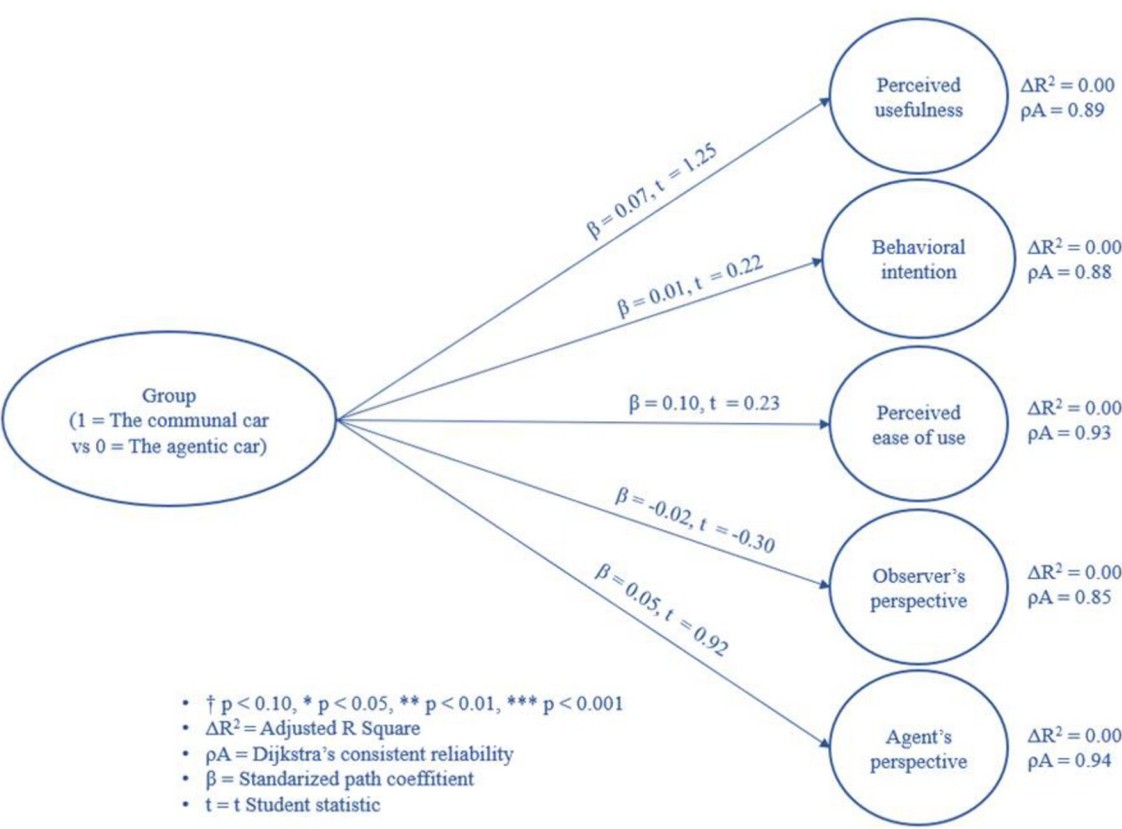

**Fig 6. Differences between conditions (agentic car vs communal one) in terms of the technology acceptance components and perspective variables (observer's and agent's).**

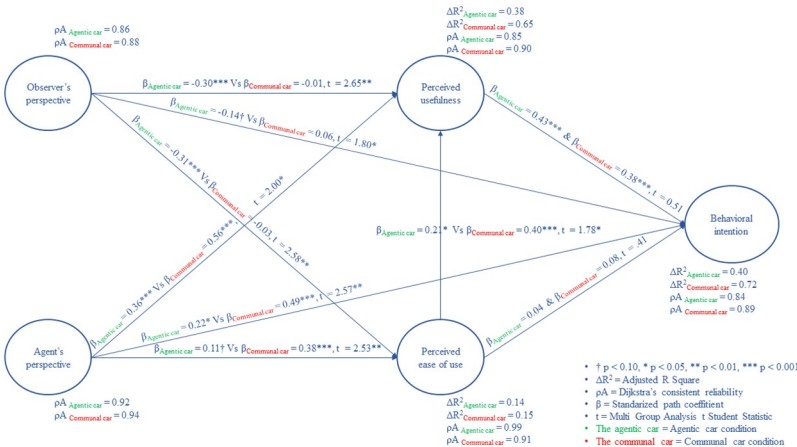

**Fig 7. Results of the conducted PLS-SEM Multi Group Analysis.**

## Analysis 2. Relations between the technology acceptance components and dual perspectives in the agentic AV condition and communal one

**Model fit.** To verify direct relationships between agent's and observer's perspectives and technology acceptance components in two different groups (agentic vs communal car advertisement), a Multi Group Analysis (MGA) was conducted [129, 130]. The MGA aims to verify differences between groups in the term of estimated path values. The model fit statistics calculated in both groups indicated similar alignment as in the previous analysis, except overall predictive GoF power [124] which was much higher. In the communal car condition, the model explained more measurements variability than in the agentic AV condition, GoF = 0.60 Vs GoF = 0.44 respectively. Both models had high predictive power, validated measurements and consistent reliabilities $\rho A > 0.86$. Model fit and quality indices for two groups are shown in Table 5. The constructs' validity is shown in Table 6.

**Multi Group Analysis results.** MGA showed that the observer's perspective was negatively related to each of the TAM components in the agentic AV condition, but in the communal AV condition these relationships were close to zero. Analysis showed also that the agent's perspective was more positively related to each of the TAM components in the communal AV

**Table 3. Zero-order correlations (top quarter), HTMT ratios (bottom quarter), and Average Variance Extracted (diagonal) between tested variables.**

| ID | Variables | 1 | 2 | 3 | 4 | 5 | 6 |
|----|-----------|-----|-----|-----|-----|-----|-----|
| 1 | Perceived ease of use | (0.87) | 0.45*** | 0.34*** | -0.27*** | 0.25*** | 0.01 |
| 2 | Perceived usefulness | 0.55*** | (0.81) | 0.64*** | -0.40*** | 0.56*** | 0.07 |
| 3 | Behavioral intention | 0.40*** | 0.83** | (0.81) | -0.36*** | 0.61*** | 0.01 |
| 4 | Observer's perspective | 0.33*** | 0.52*** | 0.48*** | (0.73) | -0.34*** | -0.02 |
| 5 | Agent's perspective | 0.28*** | 0.67*** | 0.71*** | 0.44*** | (0.86) | 0.05 |
| 6 | Group | - | - | - | - | - | (1.00) |

The bottom quarter of the table presents the HTMT ratios (good if < 0.90, best if < 0.85) with one-tailed p values (good if < 0.05); The top quarter of the table presents the zero-order correlations; Square roots of average variances extracted (AVEs) are shown on diagonal line;

* $p < 0.05$,

** $p < 0.01$,

*** $p < 0.001$.

**Table 4. Invariance analysis results.**

| Id | Factor loadings | | t (a vs b) |
|---|---|---|---|
| | AG (a) | CM (b) | |
| PEU 1 | 0.84 | 0.83 | 0.13 |
| PEU 2 | 0.86 | 0.88 | 0.19 |
| PEU 3 | 0.74 | 0.80 | 0.61 |
| PEU 4 | 0.85 | 0.85 | 0.05 |
| PU 1 | 0.77 | 0.87 | 0.97 |
| PU 2 | 0.80 | 0.86 | 0.54 |
| PU 3 | 0.73 | 0.84 | 0.06 |
| PU 4 | 0.74 | 0.82 | 0.91 |
| BI 1 | 0.71 | 0.80 | 0.97 |
| BI 2 | 0.77 | 0.80 | 0.28 |
| BI 3 | 0.81 | 0.86 | 0.45 |
| BI 4 | 0.73 | 0.86 | 0.29 |
| O 1 | 0.71 | 0.77 | 0.67 |
| O 2 | 0.76 | 0.76 | 0.06 |
| O 3 | 0.65 | 0.69 | 0.40 |
| O 4 | 0.81 | 0.81 | 0.06 |
| O 5 | 0.71 | 0.79 | 0.89 |
| A 1 | 0.83 | 0.87 | 0.45 |
| A 2 | 0.85 | 0.88 | 0.34 |
| A 3 | 0.76 | 0.81 | 0.43 |
| A 4 | 0.80 | 0.84 | 0.33 |
| A 5 | 0.73 | 0.86 | 1.36 |
| A 6 | 0.86 | 0.85 | 0.07 |

PEU = Perceived ease of use; PU = Perceived usefulness; BI = Behavioral intention; O = Observer's Perspective; A = Agent's perspective; t (a vs b) = t student ratio test for differences between experimental groups in term of the factor loadings; AG = Agentic autonomous car condition; CM = Communal autonomous car condition; * $p < 0.05$, ** $p < 0.01$, *** $p < 0.001$.

condition than in the agentic one. Further analysis showed that the perceived ease of use was more positively related to perceived usefulness in the communal AV than in the agentic one. There was no significant relation between perceived ease of use and behavioral intention in both groups because it is mediated by perceived usefulness (see moderated mediation analysis), but there was a significant relation between perceived usefulness and behavioral intention in both groups. Analysis did not detect any other significant paths and differences between conditions. Results are shown in Fig 7.

**Moderated mediation analysis.** To better understand indirect relations between tested variables in both groups, we undertook an analysis where indirect effects were moderated by groups. This moderated mediation analysis showed that the perceived usefulness significantly mediated the relation between perceived ease of use and behavioral intention in the communal car condition and agentic one. Perceived usefulness weakened relationship between perceived ease of use and behavioral intention in both conditions. Analysis showed also that the perceived ease of use and perceived usefulness mediated relationship between observer's perspective and behavioral intention in the agentic car condition, but in the communal one this effect was close to non-existent. These mediation effects were not significantly moderated by the

**Table 5. Model fit and quality indices for both groups (agentic vs. communal car).**

| Statistics | AG | CM |
|---|---|---|
| AVIF | 1.24 | 1.54 |
| AFVIF | 1.53 | 2.60 |
| GoF | 0.44 | 0.60 |
| SPR | 1.00 | 1.00 |
| SSR | 1.00 | 1.00 |
| SRMR | 0.12 | 0.09 |
| SMAR | 0.09 | 0.07 |
| $\chi^2$ | 3.01*** | 1.16*** |

AVIF = Average Variance Inflation Factor (accepted if AVIF < = 5.00, ideally AVIF < = 3.30); AFVIF = Average Full Variance Inflation Factor (accepted if AVIF < = 5.00, ideally AVIF < = 3.30); GoF = Goodness of Fit (low if GoF > = 0.10, moderate if GoF > = 0.25, high if GoF > = 0.36); SPR = Simpson's Paradox Ratio (accepted if SPR > = 0.70, ideally SPR = 1.00); SSR = Statistical Suppression Ratio (accepted if SSR > = 0.70, ideally SSR = 1.00); SRMR = Standardized Root Mean Squared Residual (accepted if SRMR < = 0.10); SMAR = Standardized Mean Absolute Residual (accepted if SMAR < = 0.10); $\chi^2$ = Chi Square; AG = Agentic autonomous car condition; CM = Communal autonomous car condition.

conditions. Further analysis showed that the perceived ease of use significantly mediated the relation between agent's perspective and the perceived usefulness in the communal AV condition, but in the agentic one this effect was close to zero. These mediation effects were significantly moderated by the experimental conditions. The last moderated mediation analysis showed that in both conditions the perceived ease of use and perceived usefulness significantly mediated relation between agent's perspective and the intention to use AV technology. These mediation effects were not significantly moderated by conditions. Results are given in Table 7.

**Table 6. Zero-order correlations (top quarter), HTMT ratios (bottom quarter), and Average Variance Extracted (diagonal) between tested variables in both groups.**

| Condition | Variables | 1 | 2 | 3 | 4 | 5 |
|---|---|---|---|---|---|---|
| AG | (1) Perceived ease of use | (0.83) | 0.39*** | 0.38*** | -0.37*** | 0.20* |
| | (2) Perceived usefulness | 0.48*** | (0.76) | 0.60*** | -0.41*** | 0.52*** |
| | (3) Behavioral intention | 0.33*** | 0.89 | (0.75) | -0.38*** | 0.44*** |
| | (4) Observer's perspective | 0.43*** | 0.68*** | 0.53*** | (0.73) | -0.25** |
| | (5) Agent's perspective | 0.22*** | 0.62*** | 0.63*** | 0.38*** | (0.80) |
| CM | (1) Perceived ease of use | (0.84) | 0.63*** | 0.52*** | -0.17* | 0.40*** |
| | (2) Perceived usefulness | 0.62*** | (0.85) | 0.79*** | -0.24** | 0.72*** |
| | (3) Behavioral intention | 0.47*** | 0.78** | (0.83) | -0.26** | 0.79*** |
| | (4) Observer's perspective | 0.23*** | 0.37*** | 0.43*** | (0.77) | -0.33*** |
| | (5) Agent's perspective | 0.34*** | 0.70*** | 0.78*** | 0.49*** | (0.86) |

The bottom quarter of the table presents the HTMT ratios (good if < 0.90, best if < 0.85) with one-tailed p values (good if < 0.05); The top quarter of the table presents the zero-order correlations; square roots of average variances extracted (AVEs) are shown on diagonal line; AG = Agentic autonomous car condition; CM = Communal autonomous car condition;

* p < 0.05,

** p < 0.01,

*** p < 0.001.

**Table 7. Moderated by car conditions (agentic vs communal autonomous car) mediation effects.**

| Indirect effects | AG (a) | | CM (b) | | Z diff (a vs b) |
|---|---|---|---|---|---|
| | β | S.E. | β | S.E. | |
| Perceived ease of use → (Perceived usefulness$_m$)→ Behavioral intention | 0.09† | 0.06 | 0.16** | 0.06 | 0.93 |
| Observer's perspective → (Perceived ease of use$_m$)→ Perceived usefulness | 0.08 | 0.06 | -0.02 | 0.06 | 0.73 |
| Observer's perspective → (Perceived ease of use$_m$ → Perceived usefulness$_m$)→ Behavioral intention | 0.14* | 0.08 | 0.00 | 0.08 | 1.26 |
| Agent's perspective → (Perceived ease of use$_m$)→ Perceived usefulness | 0.03 | 0.06 | 0.16** | 0.06 | 1.66* |
| Agent's perspective → (Perceived ease of use$_m$ → Perceived usefulness$_m$)→ Behavioral intention | 0.19* | 0.08 | 0.26*** | 0.08 | 0.67 |

β = Standardized coefficient; S.E. = standard error of the β estimate; Z diff = Z statistic estimates related to differences between experimental conditions in term of the indirect; AG = Agentic autonomous car condition; CM = Communal autonomous car condition; Variable name with the subscript $m$ = mediator;

† $p < 0.10$,

* $p < 0.05$,

** $p < 0.01$,

*** $p < 0.001$.

**General summary.** The first study showed that people, according to their declarations, raised two topics related to AVs. Firstly, they wanted to obtain information concerning the AV's communion, and secondly, they wanted to know how agentic it is. In the pilot study, two different descriptions of the autonomous CRUISE car technology were created. Results showed that the first description was perceived as much more agentic, and the second one was perceived as much more communal.

In the second study, the first analysis showed high reliability (both high ρA and AVEs) and construct discriminant validity of the used measurements (low and significant HTMT ratios). Agent's and observer's perspectives, and also TAM items, were invariant (similar factor loadings in both groups). These results indicate that the differences between experimental groups in terms of the intensity of path values can be attributed to the properties of the AV descriptions and error, rather than to properties of the variables measurement [131]. Summarizing the results of the first analysis, people in different groups rated themselves similar in terms of the two perspectives, while they also similarly accepted presented AV technology.

The second analysis in that study was focused on the acceptance of particular AV types. The agent's and observer's perspectives variables were added to predict AV acceptance components within agentic and communal AV conditions. In accordance with our predictions, observer's perspective was not negatively related with perceived ease of use, pereceived usefulness and behavioral intention of the communal AV, but these relations were negative in the agentic AV condition. This means that people who mostly focus on avoiding harm are neutral to AV technology when it is presented in a communal manner, but they do not accept it if that technology is highly agentic. Results also showed that in the communal condition the perceived ease of use was more strongly related to perceived usefulness than in agentic one. There was a significant relation between perceived usefulness and behavioral intention in both groups.

However, contrary to our expectations, agent's perspective was more strongly related to the perceived ease of use, perceived usefulness and behavioral intention of the communal AV than the agentic one. This means that people who are focused on achieving their goals by an AV are more prone to accept a communal AV rather than an agentic one. Indirect effects revealed that the perceived usefulness significantly mediated the relationship between the perceived ease of use and behavioral intention in both conditions. This means that the AV technology in both conditions was accepted, because all TAM components were positive and significant.

Analysis showed also that the perceived ease of use mediated relation between the agent's per-spective and perceived usefulness in the communal AV condition, but not in the agentic one. This significant difference may explain why people who want to achieve their goals by an AV rank higher the perceived usefulness and their intention to use the communal car in compari-son to the agentic one. It is the result of the conclusion that the former is easier to use.

## Discussion

This section is devoted to the theoretical contribution of the dual perspective model of agency and communion, and the research on the technology acceptance model, especially in regard to AVs. We also discuss practical implications for the management marketing practices in the AV industry.

### Theoretical contribution

The general objective of our investigation was to (1) determine the type of information concerning AVs that people seek (in an advertisement) and (2) identify how different AV tech-nology presentation modes (advertisements imbued with agentic or communal content) influ-ence AV acceptance among people who are either afraid of AVs or aim to achieve their goals thanks to them. The first study's findings confirm the utility of DPM-AC [25] for predicting and indicating what kind of information consumers want to acquire about AVs. According to observer's (agent's) perspective, people want to know how communal an AV is (how agentic an AV is). These results supported H1, and are consistent with a previous study [16] which reported that people expect tangible information about control issues of AVs and their safety. These results are also coherent with the notion that the communion and agency are distinctly expressed in natural language [79, 132].

The objective of the second study was to confirm how the recipient's safety concerns (observer's perspective) as well as goal orientation (agent's perspective) with regard to AV is related to technology acceptance components in different technology presentation contexts (the agentic vs communal AV advertisement). First, the pilot study provided confirmation that communal and agentic content may be utilized to create a description of AV technology [79]. People not only perceive and distinguish these types of content in themselves and others, regardless of culture [133], but also when it comes to the AV technology context.

Next, the experiment we replicated predictive properties of TAM in AV context, and also, based on DPM-AC, added new external variables (agent's and observer's perspectives), which can predict technology acceptance [41]. Nevertheless, the anticipated relationships between aagent's and observer's perspectives and TAM components vary depending on the agentic and communal AV technology advertisements. The results of the experiment showed that agentic and communal content in AV technology advertisements have moderating influence on TAM components [41] in relation to two basic perspectives [25, 91]. This means that the DPM-AC not only predicts behavior and evaluation of self and others [72, 133], but also the acceptance of AV technology [22, 41].

Our data showed that the safety concerns (observer's perspective) do not influence the acceptance of AVs in case of a communal AV technology. This pattern suggests neutrality towards AV technology acceptance. However, in case of the agentic AV technology, the safety concerns (observer's perspective) negatively influenced the AVs acceptance components. These results indicate that a communal AV does not influence AVs acceptance, but compared to agentic one, it reduces fear, anxiety and improves the overall impression made by self-con-trolled technology. We think that this pattern of results supports H2 because it is compatible with DPM-AC predictions: the more the recipient identifies with the observer's perspective,

the higher will be the acceptance of an AV technology which is more saturated with communal content than agentic one. These results are consistent with other studies where observer's mindset values communion more than agency [72, 85, 86, 134].

Our data does not support H3. In line with the DPM-AC, we predicted the more goal-oriented a person (agent's perspective) the higher an agentic AV technology acceptance is. Results showed that a higher willingness to achieve own goals by an AV (agent's perspective) was related with higher acceptance of the communal AV technology and lower acceptance of the agentic one. These results are inconsistent with DPM-AC theory and stand in opposition to H3. However, the indirect effects showed that the agent's perspective was related to higher perceived usefulness of communal AV because of its higher perceived ease of use. Our experiment did not show this pattern in the case of the agentic car condition. Therefore, according to DPM-AC the agent wants to pursue his/her goals efficiently [25, 72], and we think that the high perceived ease of use of communal AV allows for such efficiency. Nevertheless, these results are inconsistent with other studies where goal-oriented people valued agency more than communion [72, 81, 83, 87]. Based on the aforementioned self-congruity effect [90], we are convinced that in communal AV advertisement condition goal-oriented people (agent's perspective) value something very positively and this resulted in a greater acceptance.

Many previous studies underlined the role of trust in terms of the acceptance of AVs (e.g. [1, 28, 69, 135]). We believe that trust may be one of the factors which explains the higher acceptance of communal AVs over agentic AVs, even in the case of goal-oriented individuals (agent's perspective). While previous research on trust in the AV context delved into its determinants (e.g. system transparency, technical competence and situation management [48]), no study sheds light on the effect of community and emotionality on trust. However, Niu et al. [136] showed that the anthropomorphizing type of communication of AV correlates with a heightened sense of trust towards AV. Our respondents indicated the communal car as more anthropomorphized. Consequently, we assume it may be easier to trust a communal AV which is more human-like and shows feelings rather than an agentic one. Nevertheless, too great resemblance to human beings and excessive emotionality of the AV might be disadvantageous to its acceptance due to the uncanny valley effect [137, 138].

Moreover, in accordance with previous research (e.g. [66, 67]) our study supported the positive effect of perceived usefulness on the behavioral intention to use both AVs. Contrary to Hein et al. [68] as well as Lee et al. [4] and Hegner et al. [69] and, similarly to Xu et al. [29], we found impact of perceived ease of use on the behavioral intention to use AVs, but in our experiment this effect was mediated by the perceived usefulness in both conditions. While Choi and Ji [48] reported that perceived ease of use did not influence the perceived usefulness, our findings show the contrary. Perceived ease of use was related to perceived usefulness in both groups, but in the communal AV this relation was stronger than in the agentic one. Our results showed also that the explained variance of behavioral intention was high in the communal and agentic condition, respectively $R^2 = 0.72$ & $R^2 = 0.40$. These results, especially in the communal AV case, give new insights because until now existing models explained around 50% of acceptance variance [13, 28–30].

While previous research highlighted the significant role of the influence of the perceived safety of AVs on their acceptance [18, 139, 140], we took on this issue from another perspective. We confirmed that people with safety concerns (observer's perspective) were more reluctant to accept AVs than people which are goal-oriented (agent's perspective). We think that the goal-oriented people are ready to use the AV technology.

Summarizing, our study added new predictors (agent's and observer's perspectives), derived from the dual perspective model of agency and communion [25], to the traditional Technology Acceptance Models [26, 41, 61, 141]. This study also provides knowledge about

distinct acceptance properties of agentic and communal content [79, 142, 143] in technology cognition.

## Practical implications

Our investigation has implications for the promotion and communication of AVs in a congruity with agent's and observer's perspectives [87, 88, 90]. Marketers, R&D managers and engineers in the AV industry should use the social-cognitive characteristics to enhance new technology product adaptation [22, 144]. Until now, it was known that communal content rather than agentic has a stronger influence on marketing effectiveness [76, 77], because people are inclined to expect communion [145, 146]. Our study deepens this issue and suggests that both presentations of AVs have strong influence on their acceptance. Strategies based on agentic content in an AV advertisement or any form of AV-related communication will result in reluctance in the case of people who are afraid of this technology, and weak acceptance in case of people who want to achieve their goals by AVs. However, promotion and communication based on a communal content can lead to neutrality in case of fearful people, and high acceptance in case of people who want to achieve their goals by AV technology.

These insights should allow to develop more accurate communication strategies tailored specifically to the needs of AV product launches. Based on our data, industry will benefit from community-focused style in marketing. Not only high-tech companies may use those findings, but also government agencies rooting for the acceptance and diffusion of AVs due to their social benefits (e.g. safety).

## Limitations and future research

This investigation has certain drawbacks and limitations that should be pointed out. Future studies may further develop better agentic AV characteristics. In our description of the agentic car we could not incorporate every important characteristic. Various functional and agentic features may have an enormous impact on technology acceptance by recipients who want to pursue their goals by an AV (agent's perspective). Such a car would be more complete for goal-oriented people. Moreover, we used purposive sampling and recruited automobile traders and workers (sellers, distributors, brokers, and mechanics) due to their familiarity with the automobile industry. Therefore, our sample was not representative of the Polish population. Future studies should include a sample of general population, as the AV advertisements will be aimed at them.

The generalizability and precision of the findings remains to be shown by future research. Further investigations should focus on the pre-test of the TAM components which we did not control. Moreover, it is not known how manipulation changes acceptance of a technology between measurements. Adding a control group would be also beneficial, because raw relations between both perspectives and TAM components are unknown. Another possible avenue to be investigated by future research is the incorporation of the trust component into the research model in order to verify our suggestion that trust is a factor that increases the acceptance of the communal AV compared to the agentic one. This direction is promising because communion is highly related with moral judgments and preferences [70, 75, 86, 118].

Since the antecedents of technology acceptance have already been studied, the focus of researchers should be directed towards the marketing effectiveness models [147]. Studying the attitudes towards advertisements, brands, and purchase intention of AVs is crucial for a better understanding of the means of encouraging consumers to buy or rent specific AVs. Especially when the consumers are already neutral towards AV technology.

We also believe that an agentic and communal content can be combined in one advertisement. It is very important to create and verify an AV presentation that includes both types of content. Agentic content seems to strengthen community characteristics [97, 148]. We may anticipate the superior effect of such an AV presentation.

## Supporting information

**S1 Table. Latent Dirichlet allocation weights in particular topics and words.**
(DOCX)

## Author Contributions

**Conceptualization:** Konrad Hryniewicz, Tomasz Grzegorczyk.

**Data curation:** Konrad Hryniewicz.

**Formal analysis:** Konrad Hryniewicz.

**Investigation:** Konrad Hryniewicz.

**Methodology:** Konrad Hryniewicz.

**Project administration:** Konrad Hryniewicz, Tomasz Grzegorczyk.

**Supervision:** Konrad Hryniewicz.

**Validation:** Konrad Hryniewicz, Tomasz Grzegorczyk.

**Visualization:** Konrad Hryniewicz, Tomasz Grzegorczyk.

**Writing – original draft:** Konrad Hryniewicz, Tomasz Grzegorczyk.

**Writing – review & editing:** Konrad Hryniewicz, Tomasz Grzegorczyk.

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
