## [Decision Letter · Decision Letter 0]

28 Jul 2020

PONE-D-20-19621

How different autonomous vehicle presentation influences its acceptance: Is a communal car better than agentic one?

PLOS ONE

Dear Dr. Hryniewicz,

Thank you for submitting your manuscript to PLOS ONE. After careful consideration, we feel that it has merit but does not fully meet PLOS ONE’s publication criteria as it currently stands. Therefore, we invite you to submit a revised version of the manuscript that addresses the points raised during the review process.

We look forward to receiving your revised manuscript.

Kind regards,

Feng Chen

Academic Editor

PLOS ONE

Journal Requirements:

Reviewers' comments:

Reviewer's Responses to Questions

**Comments to the Author**

1. Is the manuscript technically sound, and do the data support the conclusions?

Reviewer #1: Partly

Reviewer #2: Yes

2. Has the statistical analysis been performed appropriately and rigorously? 

Reviewer #1: Yes

Reviewer #2: Yes

3. Have the authors made all data underlying the findings in their manuscript fully available?

Reviewer #1: Yes

Reviewer #2: Yes

4. Is the manuscript presented in an intelligible fashion and written in standard English?

Reviewer #1: Yes

Reviewer #2: Yes

5. Review Comments to the Author

Reviewer #1: 1. Why do the concepts agency and community in psychology represent the efficient and safe of AV technology? Was there any research support?

2. The use of some vocabulary in the manuscript is inconsistent, which may cause readers to be puzzled. For example, agent and actor express the same concept, but why should they be used confusedly?

3. In the demographic information of the questionnaire survey, the author did not consider whether participant has a driver’s license, participant’s driving experience and other factors. These factors are closely related to the participant's acceptance of AV technology and the content of their concern (safety or efficiency).

4. Questions 1/2/3 in the questionnaire of Study 1 do not seem to be significantly different. Similar questions may increase the frequency of certain words. Please explain.

5. The advertising design of agentic cars describes more details of AV technology, such as sensors, artificial intelligence and other technologies, than the advertising design of communal cars. These technologies also play a role in safety. However, in the advertising design of communal cars, the writer seems to be more inclined to anthropomorphize the car, and lacks the description of technology and the effect of technology on safety. Please explain this situation.

5. How reliable is the questionnaire to verify H2 and H3?

6. The writer mentioned that a purposeful sampling was used to recruit automobile traders and workers. This approach is also unreasonable, because these people know cars and AV technology better than the general population, and AV advertising is not aimed at this group of people.

7. Table 2 should have been cited in Analysis 1. But the writer cited Table 1.

Reviewer #2: The topic of this paper is interesting and important. The methods sound. The results are meaningful and useful. There are several suggestions to improve this paper.

1. There are some typos in this paper. For example, line 351.

2. When mentioning the effect of AVs on fuel economy, there is a review paper which should be referred.

[1] Fuel Economy in Truck Platooning: A Literature Overview and Directions for Future Research, JOURNAL OF ADVANCED TRANSPORTATION,2020, Article Number: 2604012

3. Some references have mentioned the effects of AV on the pavement sustainability. The authors need to mention this finding.

[2] Assess the Impacts of Different Autonomous Trucks’ Lateral Control Modes on Asphalt Pavement Performance. Transportation Research C: Emerging Technologies，2019, 103, 17-29.

[3] A lateral control scheme of autonomous vehicles considering pavement sustainability，Journal of Cleaner Production，Volume 256, 2020, 120669，https://doi.org/10.1016/j.jclepro.2020.120669

6. PLOS authors have the option to publish the peer review history of their article (what does this mean?). If published, this will include your full peer review and any attached files.

Reviewer #1: No

Reviewer #2: No

---

## [Author Response · Author response to Decision Letter 0]

12 Aug 2020

The answers to Reviewer’s #2 comments:

Reviewer #2: The topic of this paper is interesting and important. The methods sound. The results are meaningful and useful. There are several suggestions to improve this paper.

-- Thank you for your kind comment!

1. There are some typos in this paper. For example, line 351.

-- Typos have been corrected. 

2. When mentioning the effect of AVs on fuel economy, there is a review paper which should be referred.

[1] Fuel Economy in Truck Platooning: A Literature Overview and Directions for Future Research, JOURNAL OF ADVANCED TRANSPORTATION,2020, Article Number: 2604012

-- This article is now referenced. 

3. Some references have mentioned the effects of AV on the pavement sustainability. The authors need to mention this finding.

[2] Assess the Impacts of Different Autonomous Trucks’ Lateral Control Modes on Asphalt Pavement Performance. Transportation Research C: Emerging Technologies，2019, 103, 17-29.

[3] A lateral control scheme of autonomous vehicles considering pavement sustainability，Journal of Cleaner Production，Volume 256, 2020, 120669，https://doi.org/10.1016/j.jclepro.2020.120669

-- These articles are now referenced. 

The answers to Reviewer’s #1 comments:

Reviewer #1: 

1. Why do the concepts agency and community in psychology represent the efficient and safe of AV technology? Was there any research support?

-- We are convinced, that our article is the first which presents the link between personal dispositions and effects of the AV technology presentations. In the article we tie agency and communion with efficiency and safety because we have embedded these contents in the dual perspective theory. According to this theory, perception of the agentic and communal content (for self and others) has different adaptive value. In the article we have expanded perception of these contents on the AV technology. 

Looking back, the meaning of the above-mentioned concepts overlap in some way. There are many overlapping terms in the social cognition literature e.g. intellectually versus socially good–bad, masculinity versus femininity, instrumentality versus expressiveness, competence versus morality, dominance versus submissiveness, warmth versus competence, and trust versus autonomy, but we can still assign a common denominator of agency and community (Abele, A. E., & Wojciszke, B. (2014). Communal and agentic content in social cognition: A dual perspective model. In Advances in Experimental Social Psychology (1st ed., Vol. 50, Issue December, pp. 195–255). Elsevier Inc. https://doi.org/10.1016/B978-0-12-800284-1.00004-7)

The same may happen with terms describing the technology cognition. For example, AV technology can be named as efficient and/ or safe. However, Augmented Reality Technology can be named as have high image quality and/ or protect users' eyes. The smartphone can be fast and / or let us to communicate with other people. Nevertheless, we can still apply to the description of the properties of these technologies more general and theoretical categories of description, i.e. agency and communion. We believe that these terms can be used interchangeably. 

2. The use of some vocabulary in the manuscript is inconsistent, which may cause readers to be puzzled. For example, agent and actor express the same concept, but why should they be used confusedly?

-- In the social psychology literature, the terms “actor” and “agent” are used interchangeably, therefore, we decided to continue this practice. Of course, your opinion is very helpful because there is no need to confuse the reader. We introduced the necessary changes in the manuscript by changing the term “actor” with “agent”.

3. In the demographic information of the questionnaire survey, the author did not consider whether participant has a driver’s license, participant’s driving experience and other factors. These factors are closely related to the participant's acceptance of AV technology and the content of their concern (safety or efficiency).

-- Thank you for this comment. We considered taking into account such non-psychological factors in our model, however, these factors are already verified or mentioned by other studies:

Anderson JM, Kalra N, Stanley KD, Sorensen P, Samaras C, Oluwatola OA. Autonomous Vehicle Technology - A Guide for Policymakers. Rand. 2014. doi:10.7249/RR443-2

Hulse LM, Xie H, Galea ER. Perceptions of autonomous vehicles: Relationships with road users, risk, gender and age. Saf Sci. 2018;102: 1–13. doi:10.1016/j.ssci.2017.10.001

Liu P, Guo Q, Ren F, Wang L, Xu Z. Willingness to pay for self-driving vehicles: Influences of demographic and psychological factors. Transp Res Part C Emerg Technol. 2019;100: 306–317. doi:10.1016/j.trc.2019.01.022). 

Secondly, we decided to create a more parsimonious model. If we introduced other factors such as participants’ driving experience our model would become much more complex. Such a complex network of be difficult to interpret and would require extensive theoretical explanation. Therefore, we decided to only mention the meaning of driving experience in the theoretical sections of our paper and not to include it into the model which is still quite complex.

4. Questions 1/2/3 in the questionnaire of Study 1 do not seem to be significantly different. Similar questions may increase the frequency of certain words. Please explain.

-- We asked such similar questions because a high word frequency was needed in this study. It should be high enough for input data for topic modeling. Due to such inflation of questions, the respondents were more likely provide answers with a high number of words which can then be analyzed. In terms of the study design we were guided by classic psychometric/measurement theory where we have some similar questions which measure a concrete construct.

5. The advertising design of agentic cars describes more details of AV technology, such as sensors, artificial intelligence and other technologies, than the advertising design of communal cars. These technologies also play a role in safety. However, in the advertising design of communal cars, the writer seems to be more inclined to anthropomorphize the car, and lacks the description of technology and the effect of technology on safety. Please explain this situation.

-- Thank you for this comment. This is a very important point in our experimental manipulation. 

Introducing a technological content into the description of the communal AV would mean that the car would be perceived not only as communal but also as agentic one. My previous pilot studies (on agentic and communal product advertising) showed that introducing a soft agentic content (e.g. product is doing something or has the ability to do communal things) in a communal description creates the perception of agency and community. This pattern makes it difficult to capture the pure communion effect. 

The key to producing a proper presentation of community (as I discovered while testing numerous experimental marketing descriptions) is to embed the product as a companion in establishing and/or maintaining safe and satisfying relationships between people. In such circumstances, the respondents perceive greater community in a product. 

Nevertheless, given your suggestion, it is very important to create an AV presentation that includes both types of content. We can anticipate the superior effect of such AV presentation. An AV characterized in this way uses its agency to evoke a community, such car uses its high performance and efficiency to keep people safe. We added this notion to the limitations of the study, as well as the further direction of work.

6. How reliable is the questionnaire to verify H2 and H3?

-- These estimates are presented in the figure 7, but in the manuscript, they are mentioned as „All consistent reliabilities were acceptable ρA > 0.88”. Now we have added extra information about these reliabilities in the manuscript. „Detailed information about consistent reliabilities are showed in Fig. 7.”

7. The writer mentioned that a purposeful sampling was used to recruit automobile traders and workers. This approach is also unreasonable, because these people know cars and AV technology better than the general population, and AV advertising is not aimed at this group of people.

We agree with your comment. Students, automobile traders and workers are not the main recipients of this technology. We added a brief explanation in the manuscript and listed this issue as one of the limitations which can be answered in future research.

8. Table 2 should have been cited in Analysis 1. But the writer cited Table 1.

Thank you for this comment. That was an error, which has been corrected.

---

## [Decision Letter · Decision Letter 1]

24 Aug 2020

How different autonomous vehicle presentation influences its acceptance: Is a communal car better than agentic one?

PONE-D-20-19621R1

Dear Dr. Hryniewicz,

We’re pleased to inform you that your manuscript has been judged scientifically suitable for publication and will be formally accepted for publication once it meets all outstanding technical requirements.

Kind regards,

Feng Chen

Academic Editor

PLOS ONE

Additional Editor Comments (optional):

Reviewers' comments:

Reviewer's Responses to Questions

**Comments to the Author**

1. If the authors have adequately addressed your comments raised in a previous round of review and you feel that this manuscript is now acceptable for publication, you may indicate that here to bypass the “Comments to the Author” section, enter your conflict of interest statement in the “Confidential to Editor” section, and submit your "Accept" recommendation.

Reviewer #1: All comments have been addressed

Reviewer #2: All comments have been addressed

2. Is the manuscript technically sound, and do the data support the conclusions?

Reviewer #1: Yes

Reviewer #2: Yes

3. Has the statistical analysis been performed appropriately and rigorously? 

Reviewer #1: Yes

Reviewer #2: Yes

4. Have the authors made all data underlying the findings in their manuscript fully available?

Reviewer #1: Yes

Reviewer #2: Yes

5. Is the manuscript presented in an intelligible fashion and written in standard English?

Reviewer #1: Yes

Reviewer #2: Yes

6. Review Comments to the Author

Reviewer #1: The author gave a detailed and appropriate response to the review comments, added necessary references to the manuscript, revised and supplemented the questionable content. This paper explores the connection between personal dispositions and effects of the AV technology presentations for the first time. The topic is innovative and meaningful. The research method is complete and logical. Therefore, it is recommended to accept.

Reviewer #2: (No Response)

7. PLOS authors have the option to publish the peer review history of their article (what does this mean?). If published, this will include your full peer review and any attached files.

Reviewer #1: No

Reviewer #2: No

---

## [Editor Report · Acceptance letter]

26 Aug 2020

PONE-D-20-19621R1 

How different autonomous vehicle presentation influences its acceptance: Is a communal car better than agentic one? 

Dear Dr. Hryniewicz:

I'm pleased to inform you that your manuscript has been deemed suitable for publication in PLOS ONE. Congratulations! Your manuscript is now with our production department. 

Kind regards, 

on behalf of

Dr. Feng Chen 

Academic Editor

PLOS ONE